# Medical Care Management Based on Disaster Medicine for the Triathlon Events at the XXXII Olympiad and Tokyo 2020 Paralympic Games

**DOI:** 10.3390/ijerph20196891

**Published:** 2023-10-07

**Authors:** Masaharu Yagi, Ryoji Kasanami, Yoko Tarumi, Kenji Dohi

**Affiliations:** 1Department of Emergency, Disaster and Critical Care Medicine, School of Medicine, Showa University, Tokyo 142-8666, Japan; tarumi-y@med.showa-u.ac.jp (Y.T.); kdop@med.showa-u.ac.jp (K.D.); 2Department of Health and Physical Education, Faculty of Education, Nara University of Education, Nara 630-8301, Japan; kasanami@cc.nara-edu.ac.jp

**Keywords:** disaster medicine, heatstroke, sports, COVID-19, risk management

## Abstract

Planning the medical services for the triathlon competition at the 2020 Tokyo Olympic and Paralympic Games was predicted to be challenging because of possible last-minute changes related to the COVID-19 pandemic and abnormally high temperatures. Therefore, event planners organized and executed a disaster medical care plan, a plan for providing care during emergencies. Based on the basic medical plan for all venues provided by the Tokyo 2020 Organizing Committee, planners for the triathlon venue prepared a medical care plan according to the CSCATTT principles: Command and control, Safety, Communication, Assessment, Treatment, Triage, and Transport. After the event, planners evaluated the number of COVID-19, heatstroke, and injury cases at the venue. The events were conducted without spectators in July and August 2022 because at the last minute, planners held the event without spectators. The triathlon competition involved 638 individuals, including athletes and staff. In total, 7 cases of injuries, 3 cases of mild heatstroke, and 13 other cases were reported, with only 2 requiring emergency transportation. No cases of COVID-19 were reported from the triathlon venue, including during the observation period after the event. This medical plan was effective in preventing heatstroke and COVID-19 cases during the big event. Efficiently and effectively responding to various situations is possible in a shorter period by planning large-scale medical services for such special circumstances according to CSCATTT, a principle of disaster medical care.

## 1. Introduction

When organizing a large event, it is important to take preventive measures to prepare for possible large numbers of casualties, whether from natural disaster, mass illness, or act of terrorism, and large sports events require specific plans for sports medicine care for athletes. Amid the COVID-19 pandemic that started in late 2019, the 32nd Olympic and Paralympic Games that were originally scheduled for 2020 in Tokyo were postponed to 23 July 2021. However, the pandemic was still not under control even as the postponed date for the games drew closer; both the results of public opinion surveys and opinions of health care professionals in Japan strongly supported not holding the Games. Therefore, the debate on whether to hold or cancel the games was ongoing among Japanese government officials and the Tokyo 2020 Organizing Committee until the event. However, the committee ultimately decided to hold the Games and gave two reasons for doing so: (1) the high worldwide vaccination rates and (2) the promising effective method of protective “bubbles” to separate the general population from the Olympics athletes and support staff to prevent spread of infection. High COVID-19 infection rates persisted in Tokyo, even in the latest stages of preparation, and the emergency care system was on the brink of collapse. Many expressed concerns that the Olympics could increase the number of COVID-19 patients and further burden the emergency medical care systems; at the time, the Tokyo area was experiencing a shortage of medical facilities and human and material resources. Under these circumstances, it was difficult to ensure ample medical resources for the Olympics Games under these circumstances.

Another major challenge was protecting athletes from heatstroke because the Games would be conducted in the hottest, most humid time of the year. In addition to Japanese summers being extremely hot and humid, recent effects of global warming has accelerated the increase in the number of people who experienced heatstroke. This posed as a problem to medical care providers and to the general public and society (Figure 1) [1]. For example, 1581 people have died of heatstroke in Japan in 2018 (Figure 2) [2].

In Japan, government-led measures against COVID-19 including (1) vaccination promotion, (2) promotion of masking and disinfection, (3) prohibition of gatherings and events with large numbers of participants, (4) prohibition of international travel, and (5) detection and isolation of positive patients had been implemented, and holding the Olympics seemed to contradict the state’s own COVID-19. In addition, COVID-19 infection and heatstroke share many symptoms (e.g., fever), thereby making it extremely difficult to distinguish between them at the venue. Furthermore, building a medical system for a large-scale sporting event was a major challenge in these circumstances, and because it was the first such circumstance, it was difficult to perfectly prepare for it ad hoc.

The triathlon for which we prepared the medical care planning and management was conducted at the Odaiba Kaihin Park (Odaiba Marine Park, OMP). Because the athletes completed three consecutive events, swimming, biking, and running, the standard medical setup to prepare for heatstroke and trauma in a stadium would not be sufficient. Special medical conditions during swimming could include drowning, swimming-related pulmonary edema, and hyponatremia, and 2% of swimmers are known to require some kind of medical treatment during competition [3]. Moreover, swimming, biking, and running were performed across a vast area, including part of an ocean, which required a correspondingly extensive medical setup to cover the area.

We designed the medical plan for this triathlon to be prepared for the worst combination of circumstances—in terms of the weather and the COVID-19 pandemic—on the day of the race. As mentioned above, the special characteristics of the event were recognized by the venue medical officer (VMO), the athlete medical supervisor (AMS), and the Medical Operation Manager (MOM) recognized the special characteristics of the vent and discussed them as we formulated our disaster medicine plan [4] according to the CSCATTT principles (Table 1). For the present study, we retrospectively analyzed the validity and effectiveness of the medical plan and provide an ad hoc report of the specific plan that was implemented for the triathlon events in the 2020 Tokyo Games.

## 2. Materials and Methods

In addition to formulating a medical plan for the triathlon, we applied CSCATTT more broadly to the general medical aid plan provided by the Tokyo 2020 Games Organizing Committee. Furthermore, we made changes from day-to-day during preparations and the event itself according to the prepared plan according to plan–do–check–act. Next, we describe the tenets of CSCATTT.

### 2.1. Command and Control

Command and control refers to the organization of the disaster medicine management team. For the 2020 Tokyo Games triathlon, the international federation (IF) dispatched a medical delegate (MD), who issued the directives for the VMO to manage the medical care at the venues; the medical team organization chart is displayed in Figure 3. The teams for medical aid for the athletes and spectators were under the VMO, and a manager was assigned to each. The medical command center was managed by the VMO and the MOM and was composed of the operation staff and medical headquarters of the IF and the national federation (NF). The Tokyo Fire Department assigned a paramedic to the team to serve as a liaison to the department and support the VMO with logistics. As a doctor at an emergency medical center in the Kanto region, I was appointed as the VMO for each Tokyo 2020 venue.; the Organizing Committee felt that emergency physicians would be best suited to managing the potential medical needs at the different venues. I am a board-certified emergency medicine physician of the Japanese Association of Acute Medicine, specializing in prehospital emergency care and disaster medicine. The AMS for the triathlon, Ryoji Kasanami, is a coauthor of this paper, a physician for the Japan Triathlon Union, an orthopedic surgeon, and an expert on triathlon.

The stakeholders for the medical aid teams for athletes and for the spectators are shown in Table 2. Figure 4 shows an example of how the medical teams were laid out at the different venues. Notably, there were many people at the spectator-free event, 323 people at the Olympics and 315 people at the Paralympics in total. This included staff, athletes, coaches, referees, and other personnel who worked in the venue; VIPs such as International Olympic Committee members visited irregularly.

The AMSV was selected from the Japan Triathlon Union. The medical staff for spectators was composed of personnel dispatched from medical facilities that the VMO was affiliated with, one of whom was chosen to be the medical officer for spectators. The AMSV managed the athlete medical station only and followed the orders of the VMO rather than those of the manager of the field of play medical team. This was because the AMS was responsible for coordinating the MD and coaches, which would have made it difficult to also manage the field of play; additionally, the VMO was in the better position to gain an overall view.

### 2.2. Safety

Safety in a disaster usually refers to the physical safety of people in the area of the accident. However, for the Tokyo Games, safety did not include material hazards at the venue; it instead referred to concepts of environmental safety, including temperature, humidity, and preventive measures for COVID-19 infection.

In response to the numerous athletes affected by heatstroke at the 2019 Tokyo Olympic Triathlon event, the race was started at an earlier time to lessen the athletes’ exposure to dangerous conditions. Ice baths, drinking water, and ice for cooling were prepared in greater quantities and were provided at more stations than usual. Staff needed to wear full personal protective equipment (PPE) to prevent infection in the extreme conditions (Figure 5), which consisted of a cap, face shield, mask, gloves, and gown. Therefore, we established shifts to help the staff take more frequent breaks than normal and instructed them to regularly drink water. Furthermore, we had the entire team use a special app to manage health status rather than basing it on self-report (LINE WORKS^®^); the app was used to share positive results with the entire group of medical staff, athletes, coaches, support staff and their close contacts. Finally, all athletes, coaches, tournament management staff, and medical staff took a COVID-19 PCR test each time they entered the venue from outside the bubble.

### 2.3. Communications

Communications in a disaster refers to transmitting information effectively; information sharing among staff is critical both in general disaster medicine and in successful mass gatherings. However, we needed to keep in-person preparation meetings to a minimum according to COVID-19 social distancing guidelines. Therefore, we communicated via (1) push-to-talk over cellular, (2) LINE^®^ (an application for chat, audio, and video calls), (3) private phone, and (4) Google Meet and Zoom video meetings. The triathlon is characterized by a high rate of falls from bikes and a high incidence of severe injuries from falling, which requires accurate data to locate the injured riders; immediately identifying the location of an accident is critical to ensuring a rapid response.

However, the push-to-talk feature and the equipment approved by the Organizing Committee did not allow for accurate location by GPS; therefore, with the consent of individual motorbike team staff, we used a location-sharing app to track the riders. This gap in communications coverage occurred because the Organizing Committee and the different sports associations and medical teams involved did not prepare well enough, possibly for reasons related to the COVID-19 pandemic. Nevertheless, under the challenging conditions in 2021, flexible responses were required, and ultimately, the entire course was superimposed on a grid map; accident location information could then be shared with the command center and with referees, operation staff on the course, and field of play medical team staff (Figure 6).

### 2.4. Assessment

Assessment in disaster medicine refers to accurately evaluating the conditions on the ground at a disaster site and is an extremely essential component for planning and providing medical care under nonideal conditions. The NF, IF, VMO, AMS, and MOM held multiple meetings throughout the entire period of the 2020 Tokyo Games; before and after both the Olympics and the Paralympics, we reevaluated and modified the overall medical plan to provide the best medical care possible at each point in time. In addition, the medical staff for spectators and those for athletes met daily before and after the competition to resolve minor problems.

Throughout the entire Games, we assessed and modified our medical care plans according to the plan–do–check–adjust paradigm for identifying and resolving problems. We also conducted simulations and held ad hoc meetings with nonmedical staff to pass on relevant information for application in the next competition. For instance, we determined that our rescue plan for swimming incidents was inadequate because we were using different techniques for the competitors in the Olympic and Paralympic Games. Therefore, we held two drills to train first responders in how to retrieve casualties from the water two times. However, conducting training with a large number of participants was still a high-risk activity for COVID-19 transmission. For this reason, we filmed a video of the training and shared it with the staff so that individual staff members could do the training by themselves (Figure 7). We also conducted the ice bath trainings in multiple small groups rather than with all staff at once (Figure 8).

### 2.5. Triage

Triage is an extremely important aspect of disaster medicine for providing the best medical care possible with limited staff and medical resources and facilities; it entails assessing the urgency and severity of injuries and diseases and allocating resources to the worst cases first. At the OMP, the first medical staff member to encounter the sick or injured person performed the triage. A medical measures headquarters was set up to search for a designated medical facility destination for patient transport; however, the majority of that plan related to athletes and staff, and other patients were to be cared for within the general Tokyo Emergency Medical System.

During the Games, the COVID-19 pandemic in Tokyo was rampant, and it was predicted that further stress could collapse the medical and transport systems; we therefore determined that there would likely be challenges in providing rapid transport and treatment for Games participants in an emergency. It would have been unacceptable to conduct the event if it aggravated this collapse; therefore, it was necessary to use the general emergency medical system only moderately. Toward that aim, we established that the field of play medical staff would assess the urgency and severity of injuries and diseases in individual patients on stricter criteria than usual to determine if they could be cared for in the field of play examination room or the Olympic Village clinic. In addition, we planned routes for transporting patients that would minimize infection.

### 2.6. Treatment

Treatment in disaster medicine does not entail simply administering therapies but requires planning what types of treatment to provide, where to deploy likely limited resources (and under likely extreme circumstances), etc. As noted earlier, patients were triaged when they appeared at a facility, and we determined the appropriate venue for their care. As a general rule, we decided to respond with full PPE for care that required contact with the patient such as resuscitation measures. We used more stringent infection controls than would have been followed for previous competitions before the pandemic. Furthermore, an ambulance was prepared that was more extensively equipped and fitted with infection control and room temperature features that served as a mobile private medical emergency facility for individuals in severe distress; for instance, injured riders in the motorbike event could be located on the grid map, and their locations were communicated to the mobile emergency room. In general, individuals with injuries and illness in the running and biking portions of the event were given first aid in the athlete medical station, which was equipped with measures against infection and heat.

### 2.7. Transport

Transport in disaster medicine refers to moving injured parties to where they can be treated safely. For the 2020 Tokyo Games, we planned the transportation methods and routes considering infection control. Specifically, we placed two stand-by ambulances at the venue and four along the course (three for the mixed-gender relay).

### 2.8. Matters Examined

We examined the numbers of injuries and heatstroke cases during the Games as well as the number of positive COVID-19 PCR results during and after the Games; we also examined specific cases.

## 3. Results

The men’s individual, women’s individual, and mixed relay triathlon events were held on 26, 27, and 31 July 2021, respectively. The corresponding events for the Paralympics, Paratriathlon Standing (PTS) 4 men and PTS 2 women and Paratriathlon Visual Impaired (PTVI) for men and women were held on 28 August, and the Paratriathlon Wheelchair (PTWC) men and women and PTS5 men and women were held on the 29th. The Paralympics athletes were classed by the International Triathlon Union competition disability classification [5] (Table 3).

### 3.1. Medical Activity Cases

#### 3.1.1. Trauma Cases

At the Games, medical staff for athletes treated 12 athletes and coaches, two of whom had to be rushed to hospital due to injuries caused by falling off a bike. Medical staff for spectators treated two passers-by who were watching the event outside the venue (on the side of the road) and one event management staff member, all of whom had mild injuries or illnesses.

In the Paralympics, the athlete medical station staff only provided care for eight individuals, and all returned home without assistance; however, one of them was a PTWC athlete who visited the Olympic Village polyclinic and was diagnosed with pelvic fracture by MRI. Table 4 shows the breakdown of athletes and spectators who received care during the Games.

#### 3.1.2. Accident during Individual Women’s Race

The individual women’s race was held on 27 July, in the rain, and three athletes fell in a 5 min period during the biking. The athletes who fell first, second, and third were contacted by motorbike team physicians 2, 6, and 18 min after they fell, respectively. The player who fell first was able to walk on her own and was temporarily observed; however, a mild disturbance of consciousness was observed, so she was transported in an ambulance. The athlete who fell second had suspected shoulder dislocation and was judged to require emergency transport. The athlete who fell third did not require emergency transport but visited the polyclinic after the event and was diagnosed with ankle sprain.

### 3.2. Heatstroke

Heatstroke in the OMP venue occurred in three people during the Games, and all cases were mild. Cheering or spectating on the road was generally prohibited during the 2020 Tokyo Games. Two spectators who did not conform to these rules had heatstroke and required emergency transportation.

### 3.3. COVID-19 Infections

There were no positive COVID-19 PCR test results at the OMP triathlon venue among the Olympics and Paralympics staff, athletes, or spectators during and after the event. Among the 1,068,420 PCR tests performed throughout the entire duration of the Tokyo 2020 Games, there were only 359 positive cases, corresponding to a 0.03% incidence. The incidence of COVID-19 in the 23 wards of Tokyo before, during, and after the 2020 Tokyo Games is shown in Figure 9 [6].

## 4. Discussion

Japan is located on the Ring of Fire and is home to constant earthquakes and volcanic activity. However, the extremely frequent earthquakes are not the only natural disasters that affect the archipelago, as typhoons, torrential rains, and heavy snow also occur [7]. Japan’s Basic Act on Disaster Management addresses all stages of disaster prevention, mitigation/preparation, emergency measures, and recovery/reconstruction and stipulates that relevant public and private sectors should work together to implement various disaster countermeasures after clarifying the roles and responsibilities of the national and local governments. However, these measures are generally only designed for natural disasters such as earthquakes, tsunamis, and typhoons. The effects of global warming in recent years have increased the number of people transported by ambulance and the number of deaths due to heatstroke in Japan, seriously impacting its population.

As the Japanese government was preparing for the 2020 Tokyo Games, in addition to climate change planning, it formulated the “Heatstroke Countermeasure Action Plan”. Two components of the plan for the Tokyo Games were (1) increasing awareness of and providing information on heat such as the wet bulb globe temperature for the main venues in multiple languages and (2) cooperating with the Organizing Committee and the Tokyo metropolitan government to maximize heatstroke measures during the Games. In addition, the plan mentioned the compatibility of the dual measures against COVID-19 and heat stroke prevention. However, all aspects of medical planning for individual events, aside from information dissemination, were ultimately subsumed under the role of the VMO. We followed the CSCATTT construct to formulate a plan for providing medical care in this unique context, including accounting for the characteristics of triathlon as a sport. Following the games, we conducted a post hoc review of our plans.

Disaster medicine usually entails much higher medical demand than supply than during non-disaster times, and the Olympic and Paralympic Games are one of the greatest global events and mass gatherings in the world. Planning the emergency medical care for such an event is a particular kind of challenge, but it requires planning for conditions that are similar to a disaster. One challenge that was unique to planning for the 2020 Tokyo Games was that this was the first Olympic event in history to be conducted during a global pandemic and in abnormal climate conditions exceeding 35 °C (Figure 1). During the Games, the Japanese government still required masking both indoors and outdoors to prevent COVID-19 infection, but wearing masks obstructs heat release via breathing and was thus a concern as a factor of increased risk of heatstroke.

As the Organizing Committee was drafting this medical plan for Tokyo under these conditions, the government declared a state of emergency related to the COVID-19 pandemic in early July, which was immediately before the Games. This led to the decision on 8 August, which was just 15 days before the opening ceremony, to hold the event without any spectators, and we also revised our plan. Because there were no spectators, we reduced the number of spectator medical teams and first aid. We also canceled medical team patrols throughout the venue and moved some members of the spectator medical team to the medical command.

We attribute our success in executing a safe medical plan at the venue amid such challenging conditions to having incorporated knowledge from disaster medicine to respond to a wide range of possible scenarios. The fact that the incidences of COVID-19 and heatstroke at the OMP were lower than the rates of the general population suggests the appropriateness of the drafted medical plan. For instance, as part of the overall infection control measures for the 2020 Tokyo Games, even athletes were required to wear masks inside the venues when not competing, and PCR testing was mandatory each time they entered or left the venues from outside the bubble. The Organizing Committee received daily reports on the movements of infected people, which made it possible to identify those who were in close contact and to pay attention to their behavior. The venue was also well stocked with hand sanitizer. In addition, face shields, masks, and gloves were standard equipment for medical staff at the OMP, and gowns were worn during CPR and other concentrated contact procedures. Furthermore, medical staff were required to report their physical condition before and after work each day. We believe that our venue-specific infection control measures for the 2020 Tokyo Games as a whole minimized the number of infected people at the venues. However, if the Games had been held with spectators at the government’s discretion contrary to the advice of medical personnel, it is doubtful that we could have controlled COVID-19 infections and heatstroke cases at the venue as well as we did.

Historically, the 1920 Antwerp Games in Belgium and the 2016 Rio de Janeiro Games in Brazil were the only Olympic and Paralympic Games that were held amid global epidemics [8]. The Antwerp Games were held during the Spanish flu pandemic, immediately after World War I. However, this event occurred over 100 years ago; therefore, we were not able to find any medical articles or documentation on holding Olympic and Paralympic Games in a pandemic (PubMED, Keyword: Antwerp Olympic, Spanish flu).

The greatest merit of incorporating CSCATTT in formulating a medical plan for a large-scale event without precedence is that it allows for visualizing the worst possible situation in accordance with ironclad rules of disaster medicine. In addition, even inexperienced staff and medical volunteers can understand CSCATTT plans and actions. Triathlon competitions require special measures as the sport covers a large outdoor venue, the ocean, and spectator seats, and it is accordingly very difficult to educate staff and volunteers who are active in all places individually with the same medical plan as in ordinary competition venues. Therefore, another merit of a CSCATTT plan is that it gives all staff the same understanding and awareness of the roles they can play in a disaster, what they can expect, etc. However, CSCATTT is essentially a concept for medical care in the hyperacute phase of a disaster, and therefore, such a plan on its own lacks the detail necessary for planning medical care for large-scale events. Therefore, implementing an event-wide emergency medical care plan for the 2020 Tokyo Olympic and Paralympic Games triathlon venue required support from personnel familiar with the competition.

The Olympics and Paralympics are not just the largest sports event in the world. There are 33 events, and they take place every four years in different venues in a different country. This means that many people who are disadvantaged in a disaster or will have more difficulties adapting to environmental changes, such as foreign nationals and people with disabilities, gather from across the world. Under these circumstances, it is difficult to ensure adequate resources to address every eventuality throughout the duration of the Games. Given the circumstances, a CSCATTT-based medical aid management plan appeared to be what would be most effective for the Olympics and Paralympics.

David Jaslow et al. assert that mass gatherings are disasters and that it is necessary to create a medical plan in accordance with the 15-component mass gathering medical care plan advocated by the National Association of EMS Physicians Standards and Clinical Practice Committee [9]. Chad A. Asplund et al. advocates the following as the roles of the medical director in a medical aid plan for a triathlon during normal circumstances: (1) communication with the organizer covering all aspects of safety such as course safety; (2) development of clinical protocols; (3) recruitment and training of medical teams both at the goal and on the course; (4) organization of medical facilities including supplies for them; (5) organizing documentation protocols for athletes receiving medical care; and (6) advocating communication and coordination with EMS and local hospitals [10]. Here, we formulated a concrete medical plan by replacing these points with concepts of CSCATTT.

Next, we discuss the characteristics of injuries and illnesses during a triathlon. Triathlons in the Olympics consist of a 1.5 km swim, 40 km bike, and 10 km run. Injury rates of 17–20% per 1000 h of racing time are reported [11]; Paal K. Nilssen et al. analyzed 49,530 triathlon races and calculated an injury rate of 22.2% [12]. The incidence of injury or illness in this race was 6% lower than that in past races, However, Nilssen et al. included people up to 70 years of age, so a simple comparison with this race cannot be made.

Furthermore, in triathlons, the reported risk of traumatic injury is 1.65-fold that of general sprinting events [13]. According to Harris et al., 122 deaths occurred in the 32 years between 1985 and 2016 in triathlon events held in the USA, and 90 of these occurred during the swim, of which 93% occurred in open water [14]. The event is performed across a vast area, and medical staff must be prepared to respond to multiple injuries and illnesses that can occur at the same time. Indeed, three athletes fell from bikes in a 7 min span in the individual women’s race, all of whom required medical attention for traumatic injuries. Furthermore, heatstroke is not the only internal medical condition specific to this event; water intoxication and hyponatremia are also possible conditions that require caution. In short, managing medical aid in a triathlon requires (1) thorough understanding of triathlon as a sport and (2) thorough knowledge in traumatology, emergency medicine, and disaster medicine.

At the 2020 Games, emergency and disaster medicine specialists planned and directed medical aid as VMOs and orthopedics and triathlon sports doctors as AMSs. In addition, the fact that a MOM was in charge of the local fire department administration and emergency department administrative staff created a common understanding of disaster medical care, making it possible to coordinate effectively with the emergency medical system. Additionally, we had tested a CSCATTT medical preparedness plan for the 2019 international triathlon competition held in Japan that was the test for the Games, and we believe our previous experience with this type of plan contributed to our positive results. We recommend that any medical plan for mass gathering events be developed in accordance with CSCATTT.

## 5. Conclusions

The medical plan for the 2020 Tokyo Games triathlon competition, which was held in anomalous conditions of COVID-19 paired with abnormally high temperatures, was successfully planned from the perspective of disaster medicine. By following the principles of disaster, doctors who are familiar with emergency medicine and disaster medicine can formulate and execute effective medical plans for other anomalous circumstances such as large sporting events during a pandemic.

## Figures and Tables

**Figure 1 ijerph-20-06891-f001:**
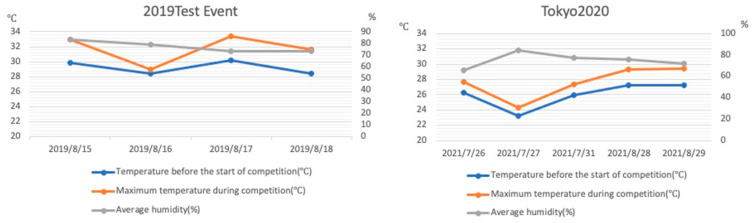
Temperatures at the Odaiba Marine Park during the 2019 test event and the Tokyo 2020 Olympic and Paralympic Games. Numerous athletes presented with heatstroke at the test event, and therefore, the race for the Games was started one hour earlier. Temperature and humidity were both lower at the start of the race at the Games.

**Figure 2 ijerph-20-06891-f002:**
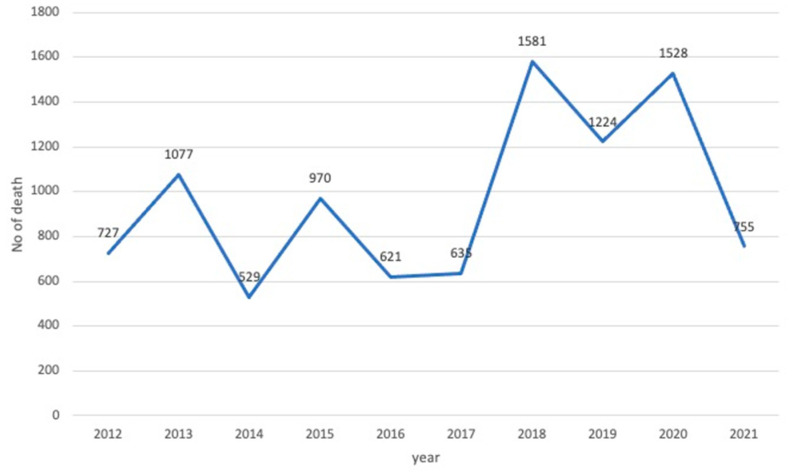
Deaths due to heatstroke in the last 10 years in Japan. In 2020, 1528 people died of heatstroke; only 755 deaths occurred in 2021, the year of the Tokyo 2020 Games.

**Figure 3 ijerph-20-06891-f003:**
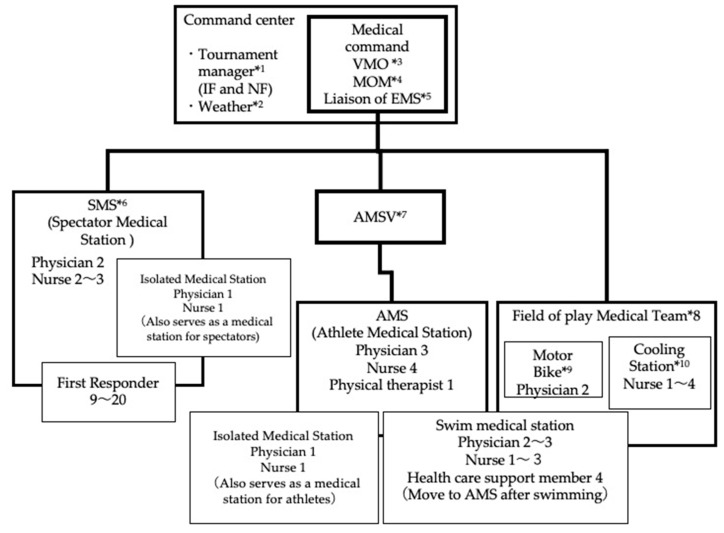
The chain of command of the Tokyo Games triathlon medical staff. Referees and representatives of the IF and NF were stationed inside the command center at the triathlon venue, managing the operations of the entire event. The medical command was composed of the VMO, the MOM, and the fire department EMS liaison. The medical staff was divided into the team for spectators and the team for athletes, and the team for athletes was further divided into the athlete medical station and field of play medical teams. The AMS managed the athlete medical stations with the IF, and the VMO directed the field of play team. *1: Tournament manager: The manager in charge of Tournament of IF and NF. *2: Weather: Person in charge of checking the weather. *3: VMO: Doctor in charge of medical supervision at the venue. *4: MOM: Administrative staff who perform support work for VMO. *5: Liaison of EMS: Coordinator with Tokyo Fire Department. *6: SMS: First aid station for injured and sick people other than athlete and coaches. *7: AMSV: Medical supervisor for athletes. *8: Field of play Medical Team: Medical team responding to injured athletes on the course. *9: Motor Bike physician: The physician rushes to the accident scene on a motorcycle. *10: Cooling Station: A waiting area for injured and retired Athletes.

**Figure 4 ijerph-20-06891-f004:**
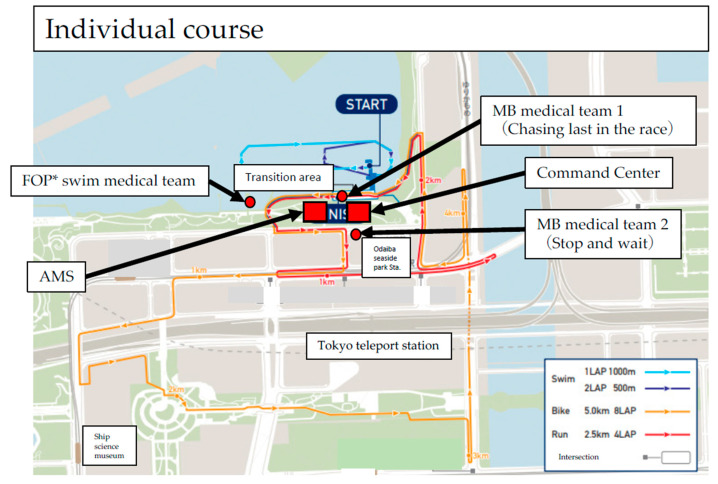
Indicates the placement of the medical team. The figure shows the individual men’s course. The VMO and MOM were placed in the command center, and the AMS and medical staff were placed in the athlete medical station. Field of play teams were assigned to respond to accidents or medical events that occurred during the swim, motorbike (MB team), and run (cooling station) events. There were two MB medical teams, one team that stood along the course and one that followed at the tail end of the race to respond to accidents. *; FOP: Field of play.

**Figure 5 ijerph-20-06891-f005:**
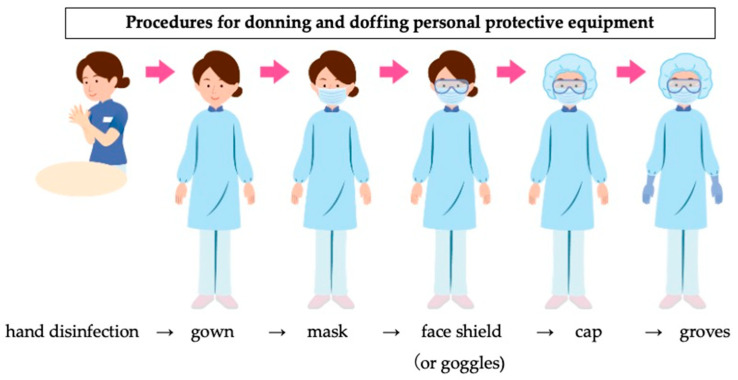
A figure showing the correct methods of wearing personal protective gear was posted to the aid stations for medical staff who needed to come into close contact with potentially infected patients.

**Figure 6 ijerph-20-06891-f006:**
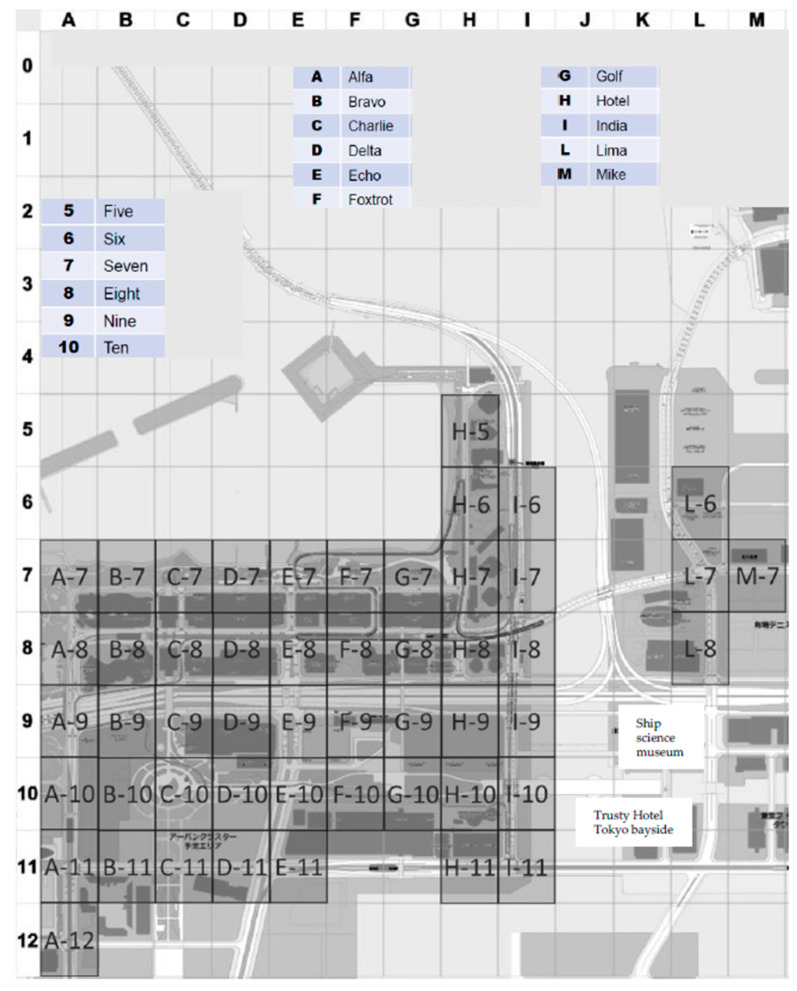
Grid map of the 2020 Olympic triathlon individual course. Accidents could be mapped on the grid and located according to their grid number, and this grid map was shared with all venue staff. A staff person who discovered an accident would report the grid number, and the medical commander would dispatch a motorbike team to the location.

**Figure 7 ijerph-20-06891-f007:**
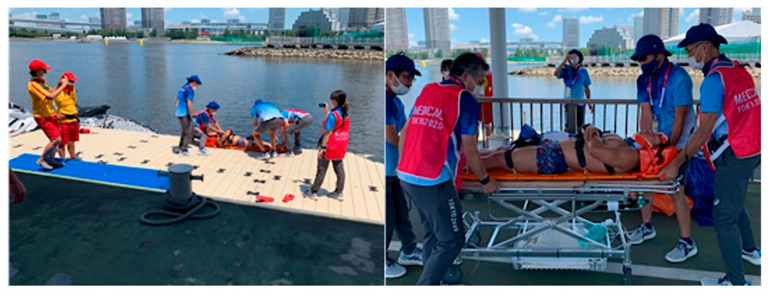
Lifting injured persons from underwater. Lifesavers transported athletes who were rescued during the swim, lifted them to the swim pontoon, performed a full spinal immobilization, and transported them to the field of play aid station. A video of the entire process was recorded for the staff who could not participate in the training.

**Figure 8 ijerph-20-06891-f008:**
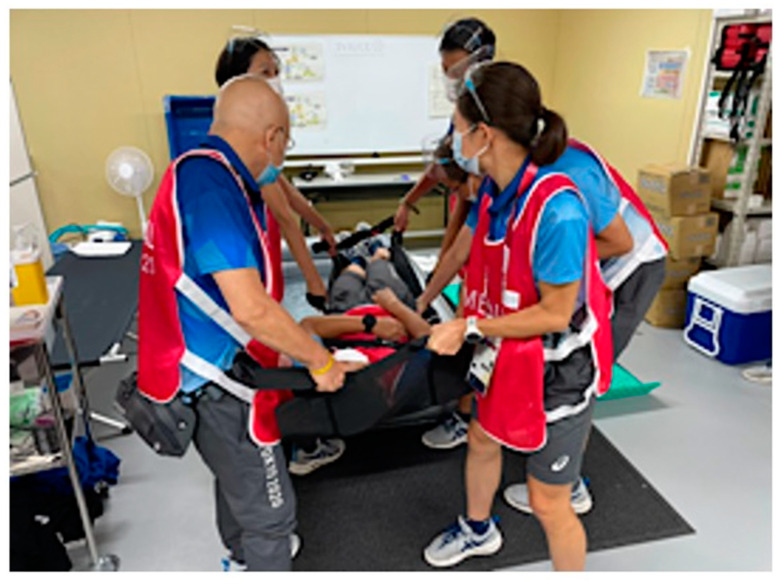
Training to put a heatstroke patient in an ice bath. Similar to the lifting training shown in Figure 7, this training was recorded on video to share with staff who could not participate in person. Ice baths are used for patients with heatstroke caused by exertion with rectal temperatures above 40 °C. Patients are promptly removed from the bath once their rectal temperature decreases to below 39 °C.

**Figure 9 ijerph-20-06891-f009:**
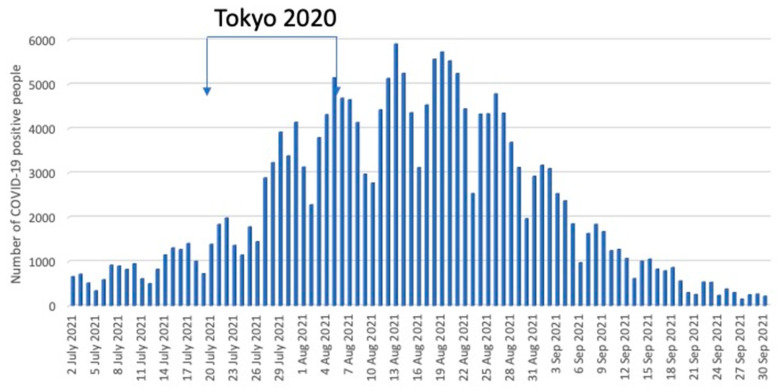
Number of COVID-19 positive people in Tokyo during the 2020 Tokyo Summer Olympics. The number began to increase before the opening of the Tokyo 2020 Games and reached the peak after it ended.

**Table 1 ijerph-20-06891-t001:** The CSCATTT principles of disaster measures. The principles were previously summarized as Major Incident Medical Management and Support.

C	Command and control
S	Safety at the scene
C	Communications
A	Assessment
T	Triage
T	Treatment
T	Transport

**Table 2 ijerph-20-06891-t002:** Stakeholders targeted for medical relief. Stakeholders at the athlete and spectator medical stations are shown. Just before the tournament, the venue was empty of spectators.

	Activity Place	Target Stakeholder
Athlete Medical	① Competition venue ▪ Athlete medical station ▪ Field of play ▪ Warmup area ② Practice venue	▪ Athlete ▪ NOC/NPC (board member) ▪ IF (referee, technical officer, etc.)
Spectator Medical	Competition venue ▪ Spectator medical station	▪ Athlete family members IOC/IPC, NOC/NPC chairman, etc. ▪ Media ▪ Marketing partner ▪ Staff ▪ Spectator

**Table 3 ijerph-20-06891-t003:** International Triathlon Union paratriathlon competition classes.

Class	Physical Disability/Condition	Degree of Disability
PTWC(PTS1)	PTWC1	Wheelchair	severelight
PTWC2
PTS2	Standing	severe 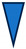 light
PTS3
PTS4
PTS5
PTVI	PTVI1	Visually Impaired	severe 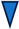 light
PTVI2
PTVI3

**Table 4 ijerph-20-06891-t004:** Numbers of triathlon athletes and event spectators who required medical intervention during the Olympic and Paralympic Games. Some athletes presented with heatstroke, but all cases were mild. Two patients required emergency transport by ambulance from the venue.

	Olympics		Paralympics	
Total number of athletes	190		80	
Medical station	AMS	SMS	AMS	SMS
Trauma (PPR%)	5 (2.6)	0	2 (2.5)	0
Heat related illness (PPR%)	1 (0.5)	0	2 (2.5)	0
Others (PPR%)	6 (3.2)	3	4 (5)	0
Emergency transfer (THR%)	2 (1.1)	0	0 (0)	0

AMS: athlete medical station; SMS: spectator medical station; PPR: patient presentation rate; THR: transport-to-hospital rate.

## Data Availability

Restrictions apply to the availability of these data. Data were obtained from the organizing committee of Tokyo 2020 and are available from the authors with the permission of the organizing committee of Tokyo 2020.

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
