# Peer review of "Medical Care Management Based on Disaster Medicine for the Triathlon Events at the XXXII Olympiad and Tokyo 2020 Paralympic Games"

_ijerph, 2023, doi:10.3390/ijerph20196891_

Round 1
Reviewer 1 Report
I read with interest this paper about medical control management of Triathlon games in Tokyo 2020.
This paper is well-written, but may be difficult to understand because of a lot of abbreviations for non-specialists. Some abbreviations seem to not be explained as IOC or only in figure 3 legend.
P5-Figure 3: as I read the figure 3, I understand that nurses and physicians can be affected to numerous stations. Is that true ? Maybe it should be explained in the text.
P6-Figure 4: the quality is poor, and the figure is impossible t read and understand. To be improved.
Author Response
Dear reviewer 1
Thank you for taking the time out of your busy schedule to review.Taking your points into consideration, I have revised it. thank you.
Masaharu Yagi

Reviewer 2 Report
attached

English is fine, one place noted where "signal" is used, did the authors mean "single"?
Author Response
Dear reviewer 2
Thank you for taking the time out of your busy schedule to review.Taking your points into consideration, I have revised it. thank you.
Masaharu Yagi

Reviewer 3 Report
Dear authors,
The manuscript titled "Planning Medical Services for the Triathlon Competition at the 2020 Tokyo Olympic and Paralympic Games: A Critical Review" discusses the challenges faced in organizing medical services for a high-profile sporting event during the COVID-19 pandemic and extreme temperatures. The authors describe how they prepared and executed a medical plan based on disaster medical care principles, particularly the CSCATTT concept. The manuscript concludes that their approach effectively prevented heatstroke and COVID-19 cases during the event.
1. The manuscript addresses an important issue of planning medical services for a major sporting event during exceptional circumstances. Given the global interest in the Tokyo 2020 Games and the unique challenges posed by the pandemic, the topic is highly relevant and significant.
2. The manuscript is well-structured, with a clear introduction, methodology, and presentation of results. The use of the CSCATTT concept is explained adequately. However, the paper could benefit from more detailed information about the specific steps taken in the medical planning process.
3. The manuscript presents data on the number of injuries, heatstroke cases, and COVID-19 cases during the event. While this data is useful, the paper lacks a comparative analysis or discussion of how these numbers relate to previous Olympic events or similar competitions. Such context would help readers understand the significance of the findings.
4. The manuscript briefly mentions last-minute changes from spectators to non-spectators, but it does not delve into the impact of these changes on the medical planning and response. Addressing the challenges and adjustments made due to these changes would provide valuable insights.
5. The manuscript suggests that the CSCATTT concept can be effective in planning for large-scale medical services during special circumstances. However, it would be beneficial to discuss the generalizability of this approach to other sporting events or disaster scenarios, as well as any potential limitations of the concept.
6. The conclusion asserts that the medical plan was effective in preventing heatstroke and COVID-19 cases. While this is a positive outcome, it would be helpful to provide more insight into the specific strategies and interventions that contributed to this success.
7. The manuscript could be strengthened by offering recommendations or lessons learned that could be applied to future events, especially considering the ongoing challenges posed by the pandemic.
The manuscript discusses an important topic but could benefit from a more detailed analysis of the data, and a discussion of the broader implications of the findings. Furthermore, providing more specifics about the planning process and the role of the CSCATTT concept would enhance the paper's value to readers involved in event planning and disaster medical care.
Author Response
Dear reviewer 3
Thank you for taking the time out of your busy schedule to review.Taking your points into consideration, I have revised it. thank you.
Masaharu Yagi
